# Outcomes of Y90 Radioembolization for Hepatocellular Carcinoma in Patients Previously Treated with Transarterial Embolization

Ken Zhao [1,*], Sam Son [1], Anita Karimi [1], Brett Marinelli [1], Joseph P. Erinjeri [1], Erica S. Alexander [1], Vlasios S. Sotirchos [1], James J. Harding [2], Kevin C. Soares [3], Etay Ziv [1], Anne Covey [1], Constantinos T. Sofocleous [1] and Hooman Yarmohammadi [1]

1   Department of Radiology, Memorial Sloan Kettering Cancer Center, New York, NY 10065, USA; erinjerj@mskcc.org (J.P.E.)
2   Department of Medicine, Memorial Sloan Kettering Cancer Center, New York, NY 10065, USA
3   Department of Surgery, Memorial Sloan Kettering Cancer Center, New York, NY 10065, USA
*   Correspondence: zhaok@mskcc.org

**Abstract:** The aim of this study was to evaluate outcomes of transarterial radioembolization (TARE) for hepatocellular carcinoma (HCC) in patients previously treated with transarterial embolization (TAE). In this retrospective study, all HCC patients who received TARE from 1/2012 to 12/2022 for treatment of residual or recurrent disease after TAE were identified. Overall survival (OS) was estimated using the Kaplan–Meier method. Univariate Cox regression was performed to determine significant predictors of OS after TARE. Twenty-one patients (median age 73.4 years, 18 male, 3 female) were included. Median dose to the perfused liver volume was 121 Gy (112–444, range), and 18/21 (85.7%) patients received 112–140 Gy. Median OS from time of HCC diagnosis was 32.9 months (19.4–61.4, 95% CI). Median OS after first TAE was 29.3 months (15.3–58.9, 95% CI). Median OS after first TARE was 10.6 months (6.8–27.0, 95% CI). ECOG performance status of 0 ($p = 0.038$), index tumor diameter < 4 cm ($p = 0.022$), and hepatic tumor burden < 25% ($p = 0.018$) were significant predictors of longer OS after TARE. TARE may provide a survival benefit for appropriately selected patients with HCC who have been previously treated with TAE.

**Keywords:** hepatocellular carcinoma; yttrium-90; radioembolization; hepatic artery embolization

## 1. Introduction

Transarterial embolization (TAE) is an accepted locoregional therapy for patients with hepatocellular carcinoma (HCC) who are not surgical candidates [1,2]. The efficacy of TAE is comparable to transarterial chemoembolization (TACE), a widely used locoregional therapy with proven survival benefits [3–5]. Advantages of TAE include its relative simplicity due to lack of pharmaceutical preparation, lack of chemotherapy-related side effects, repeatability with preservation of vascular anatomy, and approachable learning curve [6–8]. However, TAE should not be considered curative-intent therapy as disease recurrence is common [9]. Repeat TAE may be used to treat recurrent HCC, but the disease can become refractory to TAE and cease responding, a phenomenon that is well documented with TACE [10,11]. These patients require an alternative treatment. Transarterial radioembolization (TARE) with yttrium-90 ($^{90}$Y) microspheres is an alternative locoregional therapy for HCC that can be performed after other transarterial therapies. Unlike TAE and TACE, TARE's primary effect is localized delivery of high doses of radiation to the tumor, not induction of ischemia. TARE may be an effective locoregional therapy for patients previously treated with TAE.

The published experience of TARE after treatment with any transarterial therapy for HCC is limited and predominantly includes patients treated with TACE [10,12–14]. Therefore, clinical outcomes after TARE for HCC in patients previously treated with TAE

remain uncertain. The aim of this study was to evaluate the outcomes of TARE in patients with HCC who have been previously treated with TAE.

## 2. Materials and Methods

### 2.1. Ethical Statement

This is a retrospective single-institution study that was approved by the institutional review board (IRB# 16-402). Consent was waived due to the retrospective nature of the study.

### 2.2. Patient Selection

Radiology reports were searched for all patients who underwent hepatic radioembolization or mapping angiography and had a reported diagnosis of "hepatocellular carcinoma" or "HCC" between 1/1/2012 and 12/31/2022 using Montage (Nuance, Burlington, MA, USA). The electronic health record and pre-procedural multiphase contrast-enhanced cross-sectional imaging were reviewed to ensure that patients either had a tissue diagnosis of HCC or a Liver Imaging Reporting and Data System 5 lesion [15]. The procedural history and imaging of patients were reviewed for prior HCC treatment with transarterial embolization (TAE), and to confirm that the specific tumor(s) treated by TARE were previously treated with TAE. Patients who received TARE at an outside institution, did not receive TAE prior to TARE, or received TARE that targeted only untreated tumors, were excluded.

All TAE and TARE procedures were performed as part of the standard of care therapy per a consensus multidisciplinary decision between interventional radiology, hepatobiliary surgery, and medical oncology.

### 2.3. Transarterial Embolization (TAE) Procedure

The TAE procedure was performed with the patient under conscious sedation or general anesthesia at the discretion of the interventional radiologist. TAE procedures were performed using previously described technique for our institution [3,16]. Briefly, all vessels supplying the target tumor were embolized as selectively as possible with microparticles (Embosphere® Microsphere; Merit Medical, South Jordan, UT, USA). The microparticle size was per the interventional radiologist's discretion. Delivery of smaller beads was favored to achieve more distal arterial occlusion, and TAE procedures typically began with smaller size particles (40–120 μm or 100–300 μm), with progressively increasing particle size as needed to achieve stasis. Stasis was defined by contrast filling the target vessel and persisting without washout for five cardiac beats. In cases of large tumors (>10 cm diameter) with extensive vascularity or when an arterial-venous shunt was identified, larger microparticles (e.g., 300–500 μm) may be chosen as the initial particle size.

### 2.4. Transarterial Radioembolization (TARE) Procedure

A preparatory mapping angiogram was first performed to characterize tumoral arterial supply and deliver $^{99m}$Tc-MAA as per standard practice. Imaging of the chest and abdomen was subsequently performed using single photon emission computed tomography (SPECT/CT) to evaluate for extrahepatic deposition and calculate the extent of pulmonary shunting. If safe and feasible per the mapping angiogram and SPECT/CT, the patient returned for TARE typically within 14 days. All patients were treated with glass (Boston Scientific, Marlborough, MA, USA) $^{90}$Y microspheres, and the dosage of radioactivity delivered was at the discretion of the performing interventional radiologist [17,18]. Dosimetric calculations to determine the prescribed $^{90}$Y activity were performed using the medical internal radiation dose (MIRD) model for glass microspheres, in accordance with the device manufacturer's recommendations. Imaging with SPECT/CT was performed after TARE to confirm successful $^{90}$Y microsphere delivery and rule out extrahepatic deposition. Bilobar TARE was performed in two separate sessions.

## 2.5. Data Collection

Clinical characteristics, including age, gender, etiology of liver disease, Eastern Cooperative Oncology Group (ECOG) performance status, laboratory values including alpha-fetoprotein (AFP), hepatic function tests, and complete blood count, treatments for HCC including systemic therapy, surgical resection, and liver-directed locoregional therapies, date of HCC diagnosis, and date of death or loss to follow-up, were obtained by review of the electronic medical record and prior imaging. For all patients with a history of TAE and heat ablation, procedural imaging was reviewed to confirm that lesions treated with TARE had previously been treated with TAE, and that these lesions were either not treated (e.g., ablation was used to treat a different lesion) or incompletely treated with heat ablation.

Multiphase contrast-enhanced cross-sectional imaging prior to TARE, including computed tomography (CT) or magnetic resonance imaging (MRI), was reviewed to determine hepatic HCC tumor burden, including the total number of hepatic tumors, unilobar versus bilobar tumor distribution, presence of macrovascular portal vein invasion, presence of extrahepatic disease, and the largest axial dimension of the largest (index) tumor. Each patient's Barcelona Clinic Liver Cancer (BCLC) stage was determined based on a review of clinical notes, laboratory values, and imaging [19].

Procedural imaging and documentation, including reports from medical physics, were reviewed to determine the extent of the liver treated by TARE and the activity of radiation delivered. The perfused liver dose was calculated using the perfused liver volume, lung shunt fraction, and activity administered per standard single-compartment dosimetry using the MIRD model [20]. For TARE treatments with more than one radioisotope administration, the dose of the administration that covered the most tumor volume was recorded.

## 2.6. Outcome Assessment

The primary outcomes were overall survival (OS) as calculated from the time of HCC diagnosis, first TAE, and first TARE, to the time of death or last-known patient contact. Secondary outcomes were local progression-free survival (LPFS) and progression-free survival (PFS). Cross-sectional imaging was reviewed by two board-certified attending interventional radiologists (K.Z. and H.Y., with 2 and 10 years of experience, respectively).

Multiphase contrast-enhanced cross-sectional imaging after the patient's last TAE was reviewed to determine why the patient went on to receive further treatment with TARE. In particular, the first follow-up imaging study was reviewed for imaging response according to modified Response Evaluation Criteria in Solid Tumors (mRECIST), and subsequent follow-up imaging studies were reviewed for the local progression of disease [18]. The reason for further treatment with TARE was categorized as follows: TAE refractory disease (stable or progressive disease per mRECIST), residual viable disease after TAE, or local recurrence after TAE.

As per our standard clinical practice, initial post-TARE imaging was typically obtained at 1–2 months to assess initial response and rule out rapid progression or early complication. Subsequent follow-up imaging was obtained at 2–4 month intervals per the discretion of the treating interventional radiologist and the referring physician.

Follow-up multiphase cross-sectional imaging after the first TARE was reviewed to determine the best imaging response of the treated tumor(s) within 6 months according to mRECIST. An objective response was defined as a complete response or partial response.

Local progression was defined as a new or enlarging tumor detected on follow-up imaging within the portion of the liver that was treated with TARE. LPFS was defined as the time from initial TARE to local progression, death, or last-known patient contact, whichever occurred first. PFS was defined as the time from initial TARE to progression of disease anywhere in the body on follow-up imaging, death, or last-known patient contact, whichever occurred first.

Follow-up imaging, clinical notes, and laboratory results were reviewed to assess for adverse events related to TARE, which were classified using the Common Terminology

Criteria for Adverse Events v5.0 (CTCAE) [21]. Adverse events which resolved without directed treatment within 30 days of onset were defined as transient.

### 2.7. Statistical Analysis

The Kaplan–Meier method was used to determine OS from the time of HCC diagnosis, OS from the first TAE procedure, OS from the first TARE procedure, LPFS from the first TARE procedure, and PFS from the first TARE procedure. Univariate Cox regression analysis was performed on clinical and baseline imaging characteristics to determine factors associated with improved OS after TARE. The significant predictors of OS were then used to stratify patients, and the OS for each sub-group was determined using the Kaplan–Meier method. A *p*-value lower than 0.05 was considered statistically significant. Statistical analysis was performed using RStudio version 2023.06.1 Build 524 [22].

## 3. Results

### 3.1. Cohort Characteristics

This retrospective cohort included 21 patients (median age 73.4 years, 18 male, 3 female), a majority with BCLC C (71%) and a tumor burden of less than 25% of the liver (71%) (Table 1, Figure 1). Patients underwent an average of 1.71 (0.96, std. dev) TAE treatments prior to TARE (Table 2). The response to TAE and reason for treatment with TARE are summarized in Table 3.

**Table 1.** Baseline characteristics prior to radioembolization.

|  | **n = 21** |
|---|---|
| Mean age, years (range) | 74.1 (60–88) |
| Male gender, (%) | 18 (85.7) |
| Etiology, (%) | |
|     HBV | 3 (14.3) |
|     HCV | 9 (42.9) |
|     Alcohol | 3 (14.2) |
|     Other or Unknown | 6 (29.6) |
| Child–Pugh, (%) | |
|     A5 | 18 (85.7) |
|     A6 | 3 (14.3) |
| Mean MELD-Na score, (range) | 9 (7–13) |
| ALBI Grade, (%) | |
|     1 | 18 (85.7) |
|     2 | 3 (14.3) |
| Cirrhosis | 16 (76.2) |
| ECOG performance status, (%) | |
|     0 | 9 (42.9) |
|     1 | 12 (57.1) |
| BCLC classification, (%) | |
|     A | 2 (9.5) |
|     B | 4 (19.0) |
|     C | 15 (71.4) |
| Mean AFP level, ng/mL (range) | 1493 (1.1–9406) |
|     AFP $\geq$ 200 ng/mL, (%) | 9 (42.9) |
| Macrovascular portal vein invasion, (%) | 5 (23.8) |
| Extrahepatic disease, (%) | 1 (4.8) |

**Table 1.** *Cont.*

|  | n = 21 |
|---|---|
| Tumor number, (%) |  |
| 1 | 9 (42.9) |
| 2 | 3 (14.3) |
| 3 | 2 (9.5) |
| >3 | 7 (33.3) |
| Tumor distribution (%) |  |
| Unilobar | 15 (71.4) |
| Bilobar | 6 (28.6) |
| Mean index tumor diameter, cm (range) | 4.3 (2.7–9.0) |
| >4 cm, (%) | 10 (47.6) |
| ≤4 cm, (%) | 11 (52.4) |
| Hepatic Tumor Burden, (%) |  |
| <25% | 15 (71.4) |
| 25–50% | 5 (23.8) |
| 50–75% | 1 (4.8) |
| >75% | 0 |

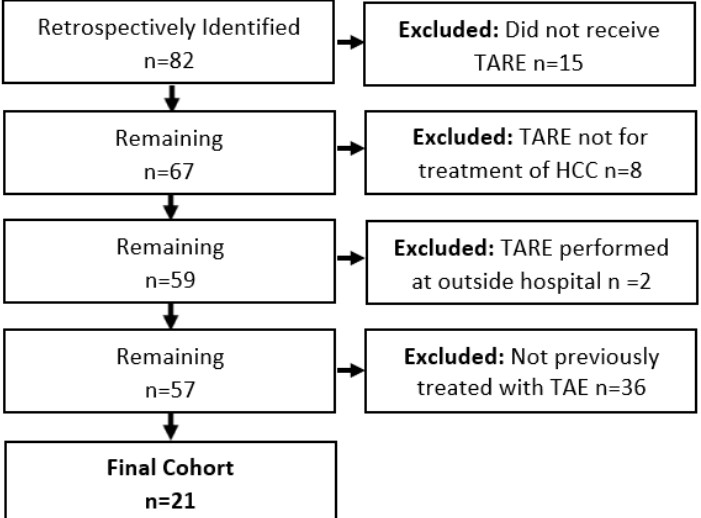

**Figure 1.** Flowchart of study cohort.

**Table 2.** Treatments other than TARE.

|  | n = 21 |
|---|---|
| Prior locoregional treatments, (%) |  |
| TAE only | 17 (81.0) |
| TAE and RFA (separate sessions) | 1 (4.8) |
| TAE and MWA (separate sessions) | 2 (9.5) |
| TAE combined with MWA (same session) | 1 (4.8) |
| Number of prior TAE treatments, per patient (%) |  |
| 1 | 12 (57.1) |
| 2 | 4 (19.0) |
| 3 | 4 (19.0) |
| 4 | 1 (4.8) |
| Systemic therapy, (%) |  |
| Prior to TARE | 2 (9.5) |
| After TARE | 13 (61.9) |
| Prior hepatic resection, (%) | 2 (9.5) |

RFA radiofrequency ablation, MWA microwave ablation.

**Table 3.** mRECIST response to final TAE and reason for further treatment with TARE.

| Response | Reason for TARE | n = 21 |
|---|---|---|
| Complete Response | Local recurrence | 4 (19.0) |
| Partial Response | Residual viable disease | 11 (52.4) |
| Stable Disease | TAE refractory disease | 3 (14.3) |
| Progressive Disease | TAE refractory disease | 3 (14.3) |

During TARE, the median perfused liver dose delivered was 121 Gy (112–444, range), and 18/21 (85.7%) patients received 112–140 Gy to the perfused liver volume.

### 3.2. Outcomes

The median OS from the time of HCC diagnosis was 32.9 months (19.4–61.4, 95% CI). The median OS from the first TAE was 29.3 months (15.3–58.9, 95% CI). The median OS from the first TARE was 10.6 months (6.8–27.0, 95% CI). Post-TARE follow-up imaging was available for 18/21 (85.7%) patients. Median LPFS after TARE was 6.3 months (3.1–18.1, 95% CI). Median PFS after TARE was 4.4 months (2.6-Not reached, 95% CI). An objective response to TARE was seen in 15/18 (83.3%) patients (Table 4).

**Table 4.** Best mRECIST imaging response within 6 months post-TARE.

| Response | n = 18 |
|---|---|
| Complete Response | 3 (16.7) |
| Partial Response | 12 (66.7) |
| Stable Disease | 2 (11.1) |
| Progressive Disease | 1 (5.6) |
| Objective Response | 15 (83.3) |

Univariate Cox analysis of cohort characteristics demonstrated that ECOG performance status of 0 ($p = 0.038$), index tumor diameter < 4 cm ($p = 0.022$), and hepatic tumor burden < 25% ($p = 0.018$) were significant predictors of better OS after TARE (Table 5). Cox analysis of perfused liver dose was not performed because of the narrow range of doses administered within the cohort. Though patients who received systemic therapy after TARE trended towards better OS, it was not a significant predictor ($p = 0.097$). The Kaplan–Meier method also demonstrated significant differences in OS after patients were stratified based on the significant predictors per Univariate Cox analysis (Table 6, Figure 2).

**Table 5.** Univariate Cox analysis of cohort characteristics as predictors of improved OS.

| Variable | Univariate Analysis | | |
|---|---|---|---|
| | HR | 95% CI | *p* |
| ECOG PS (0 vs. ≥1) | 0.299 | 0.096–0.935 | 0.038 |
| Index tumor diameter (<4 cm vs. ≥4 cm) | 0.313 | 0.117–0.847 | 0.022 |
| Hepatic tumor burden (<25% vs. ≥25%) | 0.307 | 0.109–0.862 | 0.025 |
| Patient age (<80 vs. ≥80) | 0.964 | 0.337–2.759 | 0.946 |
| AFP (ng/mL) (<200 vs. ≥200) | 0.952 | 0.362–2.5 | 0.919 |
| BCLC stage (A or B vs. C) | 0.483 | 0.177–1.316 | 0.157 |
| Child–Pugh score (5 vs. 6) | 0.962 | 0.272–3.333 | 0.947 |
| ALBI Grade (1 vs. 2) | 0.907 | 0.341–2.406 | 0.843 |
| Tumor number | | | |
| (1 vs. >1) | 0.685 | 0.257–1.852 | 0.453 |
| (≤3 vs. >3) | 0.820 | 0.301–2.222 | 0.701 |

**Table 5.** *Cont.*

| Variable | Univariate Analysis | | |
|---|---|---|---|
| | HR | 95% CI | *p* |
| Tumor distribution (Unilobar vs. Bilobar) | 0.901 | 0.313–2.632 | 0.851 |
| Macrovascular invasion (No vs. Yes) | 1.25 | 0.427–3.704 | 0.684 |
| Best response after TARE (OR vs. no OR) | 0.826 | 0.227–3.03 | 0.775 |
| Systemic therapy after TARE (Yes vs. no) | 0.392 | 0.13–1.19 | 0.097 |
| Number of prior TAE treatments (1 vs. >1) | 1.21 | 0.46–3.22 | 0.695 |

HR: hazard ratio.

**Table 6.** Kaplan–Meier OS after TARE per cohort strata.

| Variable | Median OS (95% CI), Months | *p* |
|---|---|---|
| ECOG PS | | |
| ≥1 | 9.3 (4.7-NR) | 0.03 |
| 0 | 22.3 (10.0-NR) | |
| Index tumor diameter | | |
| ≥4 cm | 5.8 (2.6-NR) | 0.016 |
| <4 cm | 22.3 (10.6-NR) | |
| Hepatic tumor burden | | |
| ≥25% | 2.8 (2.23-NR) | 0.018 |
| <25% | 18.1 (10.02-NR) | |
| Systemic therapy after TARE | | |
| No | 10.0 (3.1-NR) | 0.086 |
| Yes | 22.3 (6.8-NR) | |

NR: not reached.

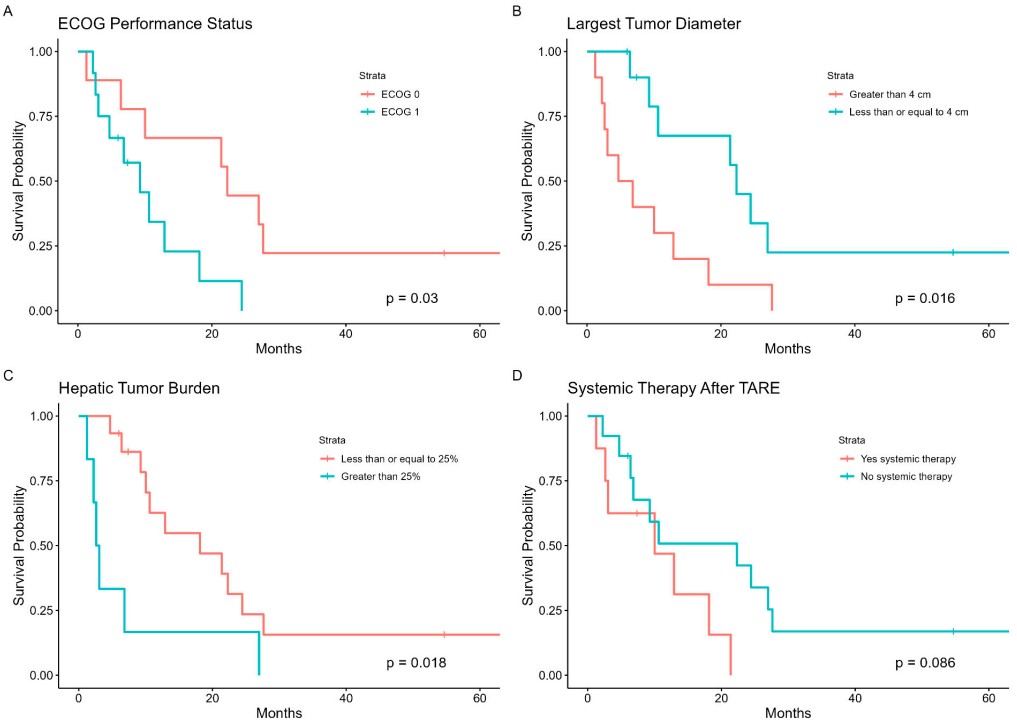

**Figure 2.** OS after TARE per cohort strata. (**A**) ECOG Performance Status; (**B**) Largest Tumor Diameter; (**C**) Hepatic Tumor Burden; (**D**) Systemic Therapy After TARE.

*3.3. Adverse Events*

Adverse events attributable to TARE are summarized in Table 7. The adverse events were predominantly grade 1–2 (34/39, 87.2%) and many (19/39, 48.7%) were transient. New ascites occurred after TARE in 6/21 (33.3%) patients, and half were symptomatic requiring paracentesis. No grade 4 or 5 adverse events occurred.

**Table 7.** Treatment-related complications by grade (CTCAE v5.0).

| Adverse Events | Grade 1–2 | Grade 3 | Transient (%) | Total |
|---|---|---|---|---|
| Total bilirubin | 3 | 0 | 2/3 (66.7) | 3 |
| AST/ALT | 4 | 2 | 5/6 (83.3) | 6 |
| Alkaline phosphatase | 3 | 0 | 1/3 (33.3) | 3 |
| Hematological toxicity (RBC, WBC, platelets, renal toxicity) | 15 | 0 | 6/15 (40.0) | 15 |
| Clinical (nausea, fatigue, abdominal pain, etc.) | 5 | 0 | 4/5 (80.0) | 5 |
| Access (hematoma, pseudoaneurysm, extravasation) | 1 | 0 | 1/1 (100) | 1 |
| Ascites | 3 | 3 * | 0/6 (0) | 6 |

AST: aspartate aminotransferase, ALT: alanine aminotransferase, RBC: red blood cells, WBC: white blood cells.
* Patients with grade 3 ascites required paracentesis for symptomatic treatment.

Three patients died within 3 months after TARE. One death was related to the marked progression of disease within the untreated contralateral hepatic lobe. The cause of death for the other two patients was neither documented within the medical record nor evident per imaging review.

## 4. Discussion

The goal of this study was to evaluate the outcomes of TARE for the treatment of HCC in patients who had previously been treated with TAE. There is a paucity of published evidence on this topic, and the available literature predominantly describes TARE for HCC in patients previously treated with TACE [10,12–14]. TAE and TACE are comparable intra-arterial locoregional therapies for intermediate-stage HCC not amenable to resection or ablation [19,23]. Both are non-curative with ischemia as the primary mechanism of tumor cell death, and it can be argued that the clinical outcomes are very similar [2–4].

The limited experience with TARE for the treatment of HCC in patients previously treated with TACE or TAE is heterogeneous. Hund et al.'s prospective study has the largest cohort, 93 patients previously treated with TACE, as well as a comparator arm of 169 TACE-naïve patients [12]. TARE was performed with resin microspheres and body surface area (BSA) dosimetry was used to calculate the dose. The median administered activity was 1.2 GBq. The estimated absorbed dose was not reported. However, a retrospective voxel-based evaluation of post-TARE Positron emission tomography-computed tomography in HCC patients treated with resin microspheres per BSA dosimetry with median delivered activity of 1.1 GBq reported a median tumor absorbed dose of 60 Gy (range 23–197) [24]. As such, it is reasonable to assume that a similar or slightly higher absorbed dose was delivered in the study by Hund et al. The median OS reported by Hund et al. was 21.5 months after TARE, which is much longer than the median OS after TARE in the current study of 10.6 months. There was also no significant difference in OS between the TACE-pretreated and TACE-naive patients. A few theories might be able to explain the longer OS. Hund et al. performed TARE with resin microparticles, which delivers more microparticles than glass and achieves a higher tumor particle density [25]. These treatment characteristics may be important in TARE for HCC in patients previously treated with embolization. In contrast, the current study utilized glass microspheres and MIRD dosimetry with an estimated median perfused liver dose of 121 Gy and 18/21 (85.7%) patients receiving 112–140 Gy. The superior OS in patients treated with TACE may be related to technical differences between the TAE and TACE procedures, including choice of embolic agent and extent of arterial stasis at the conclusion of embolization. It is also possible that differences in outcomes are artifactual from the limited sample size.

Klompenhouwer et al. reported a cohort including 30 patients previously treated with TACE, and TARE was also performed with resin microspheres. Like Hund et al., BSA dosimetry was used to determine the dose, and the average delivered activity was 1.3 GBq. Estimates of absorbed dose were not provided. Klompenhouwer et al. reported median OS of 14.8 months after TARE [13]. Similar to Hund et al., the OS reported after TARE is longer in patients treated with TACE compared to TAE in the current study.

Srinivas et al. performed TARE with glass microspheres in their cohort of 24 HCC patients previously treated with TACE or TAE, with 11/24 patients having received at least one TAE [10]. Most of their patients (83.3%) underwent segmental TARE with a mean perfused liver dose of 466.9 Gy, and the patients who underwent lobar TARE received a boosted dose with a mean dose of 190.2 Gy. This contrasts with the current study which predominantly delivered TARE with standard dosing (120 ± 20 Gy). Srinivas et al. reported a median OS of 25.7 months with the delivery of higher-perfused liver doses [10]. This is an expected result, as a dose–response relationship has been demonstrated in TARE versus HCC [26]. Partition dosimetry might be an effective strategy to achieve greater tumoral absorbed doses and improve outcomes associated with TARE for HCC in patients previously treated with embolization. However, dose escalation should be utilized cautiously, as the sensitivity of the non-tumor liver parenchyma to radiation-induced toxicity may be greater in this pretreated population. Dose–response was not evaluated in the current study due to the narrow range of doses delivered.

Wagenpfeil et al.'s multicenter retrospective study included 38 TACE-pretreated patients and a comparator group of 45 TACE-naïve patients. They reported a median OS of 13.9 months after TARE within the TACE-pretreated group, which is again longer than the current cohort. They also reported no significant difference in OS after TARE between the TACE-pretreated and TACE-naïve groups. However, the TARE treatments administered to these two groups may not be comparable. The TACE-naïve group received a mean activity of 1.71 Gbq, versus a mean activity of 1.08 Gbq for the TACE-pretreated group. Neither the method of dosimetric planning nor the utilization of glass or resin microspheres were reported.

The median OS of 29.3 months after the first TAE suggests that TARE may be associated with prolonged survival in HCC patients previously treated with TAE. The median OS after TAE for HCC is 19.6 months per a prospective randomized controlled trial previously conducted at our institution [3]. Klompenhouwer et al. reported a comparable median OS after the first TACE of 32.3 months [13]. When compared to the median OS from studies of first-line TARE, median OS after first TAE is similar to TRACE (30.2 months), similar to the personalized dosimetry arm of DOSISPHERE-01 (26.6 months), and longer than PREMIERE (18.6 months) [26–28]. It is important to note that DOSISPHERE-01 selected for patients with more advanced HCC and greater tumor burden; the personalized dosimetry arm of their cohort had a mean index tumor diameter of 10.6 cm [26]. TARE for patients previously treated with TAE may achieve a cumulative OS comparable with the best results from first-line TARE.

The median LPFS of 6.3 months suggests poorer local disease control compared to TARE as first-line treatment. TRACE reported a median time to local progression after TARE of 17.1 months and a median PFS of 11.8 months [27]. PREMIERE reported a median time-to-progression after TARE of >26 months [28]. Wagenpfeil et al. reported a median local tumor control of 6.4 months and median PFS of 4.4 months after TARE in their TACE-pretreated group, which are comparable to the current study [14]. Other studies of TARE in HCC patients previously treated with TACE or TAE did not report time to disease progression [10,12,13].

An objective response was seen in 15/18 (83.3%) patients, with most (12/18, 66.7%) patients exhibiting a partial response. Comparable studies that evaluated response with mRECIST have mixed results. Klompenhouwer et al. reported 36.7% objective response after TARE, all partial response. However, their median OS was longer than the current cohort's, despite the worse imaging response [13]. On the other hand, Srinivas et al.

reported a superior imaging response after TARE, 85% objective response inclusive of 52% complete response, and their cohort had longer median OS [10]. Within the current cohort, univariate Cox analysis did not find objective imaging response to be a significant predictor of OS. The value of imaging response per mRECIST after TARE for HCC in patients previously treated with TAE remains unclear.

ECOG performance status of 0, largest tumor diameter < 4 cm, and hepatic tumor burden < 25% were found to be significant predictors of OS after TARE. Similarly, Hund et al. found the diameter of the largest tumor < 5 cm and BCLC stage A to be amongst the strongest predictors of OS [12]. A favorable association with better performance status and lesser disease burden is typically reported with TARE for HCC [29,30].

There was a higher incidence of new ascites after TARE compared to other studies of TARE in HCC patients previously treated with TACE or TAE [10,12,13]. Klompenhouwer et al. reported that 1/30 patients developed ascites as a complication after TARE [13]. The reason for the higher incidence of ascites within the current cohort is unclear, as baseline patient characteristics are not markedly different from other comparable studies, and the incidence of laboratory and constitutional adverse events is similar [10,12,13]. The incidence of ascites may be a consequence of prior TAE or artifact from a limited sample size.

Limitations of this study include small sample size, single-institution data, retrospective design, lack of standardized procedural technique, and lack of comparative cohort. Not every patient had a fully comprehensive dataset because the information or images were not available within the electronic medical record.

## 5. Conclusions

TARE may provide a survival benefit for appropriately selected patients with HCC who have been previously treated with TAE.

**Author Contributions:** Conceptualization, K.Z. and H.Y.; data curation, K.Z., S.S. and A.K.; formal analysis, K.Z.; investigation, K.Z., S.S., A.K. and H.Y.; methodology, K.Z. and H.Y.; software, K.Z. and J.P.E.; supervision, K.Z. and H.Y.; visualization, K.Z. and S.S.; writing—original draft preparation, K.Z.; writing—review and editing, K.Z., B.M., J.P.E., E.S.A., V.S.S., J.J.H., K.C.S., E.Z., A.C., C.T.S. and H.Y. All authors have read and agreed to the published version of the manuscript.

**Funding:** Funded in part through the NIH/NCI Cancer Center Support Grant P30 CA008748.

**Institutional Review Board Statement:** The study was conducted in accordance with the Declaration of Helsinki, and approved by the Institutional Review Board of Memorial Sloan Kettering Cancer Center (IRB# 16-402, approved 5/3/2016).

**Informed Consent Statement:** Consent was waived due to the retrospective nature of the study.

**Data Availability Statement:** The data presented in this study is available in this article.

**Acknowledgments:** Memorial Sloan Kettering Cancer Center, requires that all peer-reviewed research acknowledge the Cancer Center Support Grant in the funding acknowledgements, be deposited in PubMed Central (PMC) and assigned a PMCID, and properly associate Selwyn Vickers with the P30 Core Grant in MyNCBI/My Bibliography. Of note, Vickers is not an author on this paper.

**Conflicts of Interest:** Dr. Zhao has received a research grant from the SIR Foundation. Dr. Erinjeri is a consultant for AstraZeneca. Dr. Alexander is a consultant for Boston Scientific. Dr. Harding has received research support from NCI P30-CA008748, NCI U01 CA238444 04, the Society of Memorial Sloan Kettering Cancer Center, Experimental Therapeutics Center, and Cycle for Survival, AbbVie, Bristol Myers Squibb, Boehringer Ingelheim, CytomX, Debiopharm, Eli Lilly, Genoscience, Incyte, Kinnate Biopharma, Loxo @ Lilly, Novartis, Polaris, Pfizer, Tvardi, Zymeworks, Yiviva. Dr. Harding is a consultant for Adaptimmune, AstraZeneca, Bristol Myers Squibb, Exelexis, Elevar, Eisai, Genoscience (uncompensated), Hepion, Imvax, Merck (DSMB) Medivir, QED, RayzeBio, Servier, Tempus, Tyra, and Zymeworks (uncompensated). Dr. Ziv has received research grants from MSK, Druckenmiller, NETRF, AACR, NANETS, SIR, RSNA, Ethicon, Novartis. Dr. Covey is a stockholder of Amgen, a member of the advisory board of Boston Scientific, a paid speaker for Oncology Live, and section editor of AJR. Dr. Sofocleous has received research support from the NCI/NIH, SIO, SIR

Foundation, Boston Scientific, MIM, SIRTEX, Johnson and Johnson and VARIAN. Dr. Sofocleous is a consultant or member of the advisory board for Terumo, Varian, SIRTEX, and Johnson and Johnson. Dr. Sofocleous has served as a member of the Board of directors of SIO 2017–2023 and is currently a member of the SIR executive council. Dr. Yarmohammadi received research grants from the Thompson Foundation and Guerbet. The remaining authors declare no conflict of interest. The funders had no role in the design of the study; in the collection, analyses, or interpretation of data; in the writing of the manuscript; or in the decision to publish the results.

## Abbreviations

**AFP:** Alpha-fetoprotein, **ALBI:** Albumin-bilirubin, **ALT:** alanine aminotransferase, **AST:** aspartate aminotransferase, **BCLC:** Barcelona Clinic Liver Cancer, **BSA:** Body surface area, **CT:** Computed tomography, **CTCAE:** Common Terminology Criteria for Adverse Events, **ECOG:** Eastern Cooperative Oncology Group, **HCC:** Hepatocellular carcinoma, **HR:** Hazard Ratio, **LPFS:** Local progression free survival, **MIRD:** Medical internal radiation dose, **mRECIST:** modified Response Evaluation Criteria in Solid Tumors, **MRI:** Magnetic resonance imaging, **NR:** Not reached, **OS:** Overall survival, **PFS:** Progression free survival, **RBC:** red blood cells, **SPECT/CT:** Single photon emission computed tomography, **TACE:** Transarterial chemoembolization, **TAE:** Transarterial embolization, **TARE:** Transarterial radioembolization, **WBC:** white blood cells.

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
