# Peer review of "Outcomes of Y90 Radioembolization for Hepatocellular Carcinoma in Patients Previously Treated with Transarterial Embolization"

_curroncol, doi:10.3390/curroncol31050200_

Round 1

Reviewer 1 Report

Comments and Suggestions for Authors

Please see attached word document for more details. 

-          Page 1, line 30: Safety and Efficacy of TAE is comparable to TACE in certain clinical scenarios. I wouldn’t feel comfortable to say that it is equivalent (as in equal to TACE). The authors cited Dr. Brown’s clinical trial (reference 3), which itself alludes that many studies have shown differences in safety and efficacy between TAE and TACE (i.e. the data remains heterogenous). Additionally, the authors state in Page 1 line 35 that “TAE should not be considered curative-intent therapy as disease recurrence is typical”. Please rephrase sentence accordingly.

-          Suggestion: In the introduction, it would be beneficial to devote two or three sentences as to why/when one would do TARE post any transarterial therapy. I suspect that easier thing to say would be in the setting of a mass abutting the colon/duodenum, so one treats it with TAE, per say, and then there may be residual viable disease that can then be safely treated with TARE. Anyway, providing the reader with an example of a clinical scenario when one would see TARE after TAE would be helpful.  

-          Dosimetry mention in the method section is lacking more detailed information. Yes, MIRD model was used, but what Gy dose to the angiosome was prescribed in general? Was personalized dosimetry utilized for all cases? Please provide more specific information as to the doses prescribed so as to provide color for the reader to know the TARE doses given within this study.

-          For Table 1, please provide the median and IRQ for the number of TAE procedures before TARE to provide a more nuanced understanding of the patient demographic. Average is helpful but can be somewhat misleading.

-          Table 1 “Tumor Number”: Does this refer to the number of tumors in the liver or the number of tumors that underwent TAE and then TARE? Please clarify

-          Figure 1: Please elaborate  what the 82 retrospective identified means? Was this the number of patients with HCC that underwent MAA? Method say HCC patients that underwent TARE were identified, but yet the figure 1 says 15 patients did not receive TARE? Please clarify.

-          Table 2: Please clarify in the results section text if RFA/MWA occurred before TAE or after TAE. Consider updating the table to reflect this (for example, for patients that underwent RFA and then TAE, can consider RFA + TAE. Vs for patients that underwent TAE and then RFA, can consider TAE + RFA). On this note, (if applicable) please provide satisfactory rationale for keeping patients that underwent RFA after TAE for analysis in this study.

-          Minor point: As authors know, some systemic therapies (Atezo/Bev for example) can make angiography very difficult and can at times significantly alter the vascular anatomy (making TARE hard, if not unsafe to perform). Can the authors comment, perhaps in the discussion section, whether systemic therapies utilized after TAE impacted their ability to perform TARE?

-          Page 5, line 168: Dosimetry information is significant lacking. Was 121 Gy to the angiosome? Does 85% of Patients receiving 112-140 Gy include liver + lung doses? Information is incomplete/unclear. Please provide more clarity to dose to angiosome, and (if possible) dose to tumor received.

-          Since method section stated that SPECT/CT was performed, please provide information on Lung Shunt Fraction before TARE and with TARE. It is of interest to know if TAE would significantly reduce LSF (for example, instead of the usual <5% LSF, was <1% LSF seen for MAA and TARE LSF?)

-          Page 6 line 171 says “Median OS from first TARE” suggesting that subsequent TAREs were completes. Can the authors confirm that the data presented is only for the first TARE and not the combination of all TARES?

-          Page 6 line 181: “Tumor absorbed dose was not analyzed because of the narrow range of doses administered within the cohort.” Sure – we can understand not providing tumor absorbed dose (we are aware it can take a bit of work to get this). But to make this statement the authors need to at least provide more information on the prescribed dose to the angiosome. In addition to this, please provide dose received to the angiosome. Dose to the angiosome (and by extension the tumor) is very very important, so it is critical that more dosimetry information is provided. 

-          Please comment if there was any patients who had biliary injury after TAE. If so, did the author’s change their TARE dosing or post-TARE medical management?

-          Please clarify in the methods section what constituted as ‘transient’ Adverse Event. How many weeks vs months had to pass until the change was considered transient?

-          Particle loading is brought up in the discussion – interesting hypothesis as to why the OS in this data cohort is less than in previously mentioned studies. More interestingly, the previously mentioned study (Hund et. Al.) had a tumor mean dose of roughly 60 Gy with resin particles – which is a relatively low tumor mean dose (but not unheard of given their now outdated use of BSA). It is known that resin and glass particles have different tumor dose response thresholds likely based on differences in particle deposition and composition (with resin having slightly lower tumor dose response threshold than Glass microspheres).  With that in mind and the limited dosimetry information given in this study, it is suggested by this study’s authors that the mean tumor dose for this study is likely to be >121 Gy (because the mean absorbed dose (supposedly to the angiosome) was reported as 121 Gy and (one assumes the TNR for most of the treated tumors in this study are likely >1). As such, the authors are implying the even though they likely gave a higher Gy to the tumor, their study’s OS was less than that reported by Hund et al may have been because of lower particle loading differences given in this study. This suggests two areas of improvement in this study: A) Provide more detailed dosimetry information. B) consider providing particle loading information for this study.  I would expect point A to be done so that this paper can be fully accepted. Point B can be a bit hard to do (I’ve done it for my papers, so I understand), so I think it would be okay to publish this paper without the particle loading information assuming the other reviewers said the same.

-          Page 8 line 231: Please elaborate on what do you mean when you say different TAE techniques may have affected the OS.

-          The authors mention that their complete response rate is lower than other reported similar studies. Again, the studies mentioned by the authors (such as Srinivas study) had utilization of more personalized dosimetry (They used MIRD >150 Gy to lobar treatments and >205 Gy for segmentectomies). As such, this again highlights the importance of beefing up the dosimetry section of this study.

Reviewer 2 Report

Comments and Suggestions for Authors

This article is well written but it contains small problems (see below). Thus, it is not acceptable for publication in the present form.

Major points

1)                I agree with the authors that TARE may provide a survival benefit for selected patients with HCC previously treated with TAE. I cannot understand accurately why TAE (not TACE) was performed in their hospital (because of its simplicity? its repeatability?).

In my opinion, the authors should compare the results with previous reports describing TARE after TACE more deeply in Discussion.

2)                Materials and Methods 2.3-2.4 TAE procedure and TARE procedure:

In my impression, there were no strict procedures (most are “per the discretion of the interventional radiologist). Please mention this problem briefly in Discussion.

3)             Results 3.1. Cohort characteristics: All cases were cirrhotic?

4)             3.2-3.3. Outcomes and Adverse events: Not age-related?

5)       Discussion: Please describe what lessons this study has suggested (regarding     treatment characteristics).

Minor points

              This article included many abbreviations. Please add a list of abbreviations. It will help readers understand the content more easily.

Round 2

Reviewer 1 Report

Comments and Suggestions for Authors

Thank you for the response to the comments and revisions to the manuscript. Overall, I think this study contributes nicely to the relatively small amount of literature on the matter. Would recommend authors to consider stronger and more detailed tumor dosimetry and particle loading sections in their future studies - as this is now a very 'hot topic' in Y90 literature.